# Knowledge, Attitudes and Perception of Mosquito Control in Different Citizenship Regimes within and Surrounding the Malakasa Open Accommodation Refugee Camp in Athens, Greece

**DOI:** 10.3390/ijerph192416900

**Published:** 2022-12-16

**Authors:** Antonios Kolimenakis, Demetrios Tsesmelis, Clive Richardson, Georgios Balatsos, Panagiotis G. Milonas, Angeliki Stefopoulou, Olaf Horstick, Laith Yakob, Dimitrios P. Papachristos, Antonios Michaelakis

**Affiliations:** 1Scientific Directorate of Entomology and Agricultural Zoology, Benaki Phytopathological Institute, 145 61 Kifisia, Greece; 2School of Applied Arts and Sustainable Design, Hellenic Open University, 263 35 Patra, Greece; 3Department of Economic and Regional Development, Panteion University of Social and Political Sciences, 176 71 Athina, Greece; 4Research to Practice Group, Heidelberg Institute of Global Health, Heidelberg University, 69120 Heidelberg, Germany; 5Department of Disease Control, Faculty of Infectious & Tropical Diseases, London School of Hygiene and Tropical Medicine, London WC1E 7HT, UK

**Keywords:** mosquito borne diseases, KAP, migrants’ refugees’ health, mosquito control, public health, vectors

## Abstract

The study aims to evaluate the Knowledge, Attitude and Perception (KAP) of different societal groups concerning the implementation of targeted community-based mosquito surveillance and control interventions in different citizenship regimes. Targeted surveys were carried out within Malakasa camp for migrants and refugees, neighboring residential areas and urban areas in the wider Athens metropolitan area to investigate different knowledge levels and the role that both local and migrant communities can play in the implementation of community-based interventions based on their attitudes and perceptions. A scoring system was used to rate the collected responses. Results indicate different levels of KAP among the various groups of respondents and different priorities that should be considered in the design and execution of community interventions. Findings indicate a lower level of Knowledge Attitudes and Perceptions for the migrants, while the rate of correct answers for Perception significantly improved for migrants following a small-scale information session. The study highlights disparities in the levels of knowledge for certain public health issues and the feasibility of certain approaches for alleviating health-related challenges such as mosquito-borne diseases. Findings suggest that essential preparedness is needed by public authorities to respond to public health challenges related to migration and the spread of vector-borne diseases.

## 1. Introduction

According to WHO [1], the need for a comprehensive approach to insect vector control to counter the impact of vector-borne diseases has never been more urgent. Under the strain of several challenges, including financial crisis, escalating poverty, and mass migration, Southern Europe has become a region particularly vulnerable to the introduction and spread of invasive mosquito vectors, such as *Aedes albopictus*. A complex operation of the *Aedes* mosquito species is depending on strategic integrated vector management; apart from the usual larvicidal spraying activities, it often requires active involvement of citizens, which is hard to achieve [2,3]. Under different socio-economic pressures, including mass migration, it is imperative that communities and governments prepare their response to infectious disease threats by taking into consideration additional challenges associated with these settings [4].

The introduction of invasive mosquito species into the Mediterranean region poses new challenges for both scientists and policy makers. The last decade has seen the widespread invasion of the Asian tiger mosquito (*Ae. albopictus*) in various urban ecosystems of Southern Europe [5], in addition to the existing threat of the spread to this region of *Ae. aegypti* (already established in Georgia and northeastern Turkey). In Europe, *Aedes*-borne diseases have lately emerged or re-emerged as a public health threat transmitted by the invasive *Aedes* species (i.e., *Ae. albopictus* and *Ae. aegypti*). In more detail, in Madeira (Portugal), after the invasion of *Ae. aegypti* in 2005, an outbreak of dengue with more than 2000 cases was reported in 2012 [6]. On the other hand, the establishment of *Ae. Albopictus* in many European countries has caused several outbreaks; in Italy, in 2007 and 2017, there were two large outbreaks of chikungunya [7,8], while there have been sporadic cases of autochthonous dengue and chikungunya in Croatia, France, Italy and Spain [9,10,11,12]. Recently, *Ae. aegypti* has been recorded in Cyprus, close to Larnaka’s airport. This fact, in correlation with the increasing number of autochthonous transmissions of dengue virus in mainland France in 2022, appears to be an alarm for all the European and Mediterranean countries [13].

In Greece, since 2010, several outbreaks of West Nile virus infection related with *Culex* mosquito species have been recorded. Furthermore, since 2009 several malaria cases have been recorded, mainly as sporadic introduced cases, and in 2011–2012 in clusters [14]. Up to date, there is no previous experience of autochthonous mosquito–borne diseases related to *Aedes* species, such as Dengue, Zika and Chikungunya, in the Greek territory.

The spread of new epidemics and their overall socio-economic consequences remain largely unpredictable and extremely difficult to evaluate [15]. Additionally, mass migration, especially to borderlands and urban areas with inadequate infrastructure, poses even greater challenges to public health policy makers [16,17]. Poverty and other related social determinants, together with unequal access to health services in host countries, increase vulnerability to infections and decrease the opportunities for diagnosis and treatment [18,19].

An array of advanced vector control methods and tools is currently being considered globally. These include both community engagement [20,21] and technologically based approaches such as the sterile insect technique, genetically modified mosquitoes, Wolbachia-infected mosquitoes and citizen science-based platforms [22,23]. While these approaches can potentially help to limit the transmission of diseases such as malaria, dengue, Zika and yellow fever, they pose a collection of ethical, regulatory and social questions, such as public acceptance, socio-economic welfare, cost-effectiveness and efficacy rates of application, that require further investigation to assess the feasibility and added value of their implementation within modern societies.

The aim of the current study is to evaluate the Knowledge, Attitudes and Perceptions (KAP) of different societal groups concerning the implementation of targeted community-based mosquito surveillance and control interventions under different citizenship regimes [24]. Based on a series of surveys covering these groups, the study investigates the different knowledge levels as well as the role that both local and migrant communities can play in the implementation and supervision of these community-based interventions. The overall aim was to evaluate the feasibility of community-driven interventions under different modes of citizenship. The current study builds on previous methodological frameworks and findings of KAP surveys conducted in the southern Mediterranean area, where community-based mosquito vector control approaches are already being implemented and tested [25,26].

## 2. Materials and Methods

### 2.1. Study Areas and Samples

The questionnaire survey of migrants and refugees was carried out in the Open Accommodation Refugee Camp at Malakasa, located on the geographical border of the Athens area (Figure 1). There were 264 prefabricated accommodation containers and 29 apartments each hosting two families in separate rooms with common Water, Sanitation and Hygiene (WASH) Facilities (shower/bath), while a number of newly arrived migrants resided on temporary settlements including tents. Also, the Hellenic Water Company (EYDAP) provides untreated water and suitable treatment systems have been installed in order to comply with the relevant EU and national standards. Further surveys, using essentially the same questionnaire, were carried out among citizens of the Municipality of Oropos, which is immediately adjacent to the Malakasa Camp, and the Municipality of Moschato-Tavros which is an urban municipality closer to the centre of the Athens metropolitan area, with a population of 40,413 inhabitants [27] and a total area coverage of 52,400 hectares. While Moschato-Tavros is heavily built-up, Oropos is less densely populated, with a population of 52,300 inhabitants [27], and a total area coverage of 3,174,600 hectares, with many of the houses being weekend and holiday homes of residents of the city.

### 2.2. Interviewing Process

Interviews using structured questionnaires were conducted from June 2020 to October 2020, initially in Malakasa and subsequently in the areas of Oropos and Moschato-Tavros municipalities. Most of the camp residents are Afghan nationals and interviews were conducted in their language by an interviewer accompanied by a translator who was provided by the International Organization of Migration office operating within the camp. Interviews were conducted on the camp premises, mostly outside the respondent’s residence, and the sample was selected based on the availability and willingness of each questioned subject to participate in the interview. Interviews with citizens of Oropos and Moschato-Tavros were conducted by a group of up to three interviewers who approached potential participants in public places of each municipality. Interviews in all three locations lasted approximately 5 to 10 min. One hundred interviews were carried out in Malakasa camp, 50 interviews in Oropos municipality and 159 in the municipality of Moschato-Tavros.

A second round of interviewing was carried out in the migrant camp, to test further the migrants’ perception of their personal responsibility for dealing with the mosquito problem, utilizing the same interviewing approach as in the first round. Interviews in this round were preceded by a brief information session on the “correct answers” of the first KAP questionnaire. Participants were also asked whether they had participated in the first round of interviews. Then the question about who was responsible for dealing with the mosquito problem was repeated. A total of 80 questionnaires were collected during the second round, while almost 80% of the respondents had not participated in the previous round of interviews. The overall structure of the interviewing process is presented in Figure 2.

### 2.3. Questionnaire

The structured questionnaire was designed on the basis of a small-scale pilot study by personal interviews among migrants and refugees residing in the camp. The structure of the questionnaire was based on previous similar conducted studies in the Greek territory [25,28], which was adapted to the needs of this survey by being pilot-tested for its efficiency and feasibility within the camp’s context. Based on the pilot test, conducted to a small sample within the Malakasa camp with the support of International Organization of Migration office translators, the final set of questions was determined. It included questions divided into three sections intended to evaluate Knowledge, Attitudes and Practices, respectively. The core questions in these sections were the same for all three groups while questions on the respondents’ background were differentiated in order to capture different residence characteristics and demographic information of each group. A final question asked who the respondents considered to be responsible for dealing with the mosquito problem: “the Camp authorities” (or “State/municipal authorities” in Oropos and Tavros), “themselves” or “both”.

For all study areas the same system for scoring knowledge, attitudes and perceptions was used as in previous studies conducted in Greece [25,28]. In terms of the statistical analysis employed, the percentage distributions of responses to individual questionnaire items, and knowledge and attitude scores, were compared between the three samples using chi-squared tests.

### 2.4. Knowledge Score

The knowledge score of respondents was based on three questions that aimed to evaluate their general knowledge about mosquitoes. The first question asked respondents which mosquitoes bite, females or males. Correct answers were scored 1 and wrong answers 0. The second question asked respondents their knowledge of mosquito breeding loci (stagnant water, soil, drains, mud, plant saucers, waste and tires, plants, running water, ponds, flowerpots) and 1 point was scored if they could identify at least one location correctly without selecting any incorrect option, otherwise 0. A third question asked which among a list of diseases (dengue, measles, malaria, chikungunya, AIDS, tuberculosis, bird flu, Zika, West Nile virus) are transmitted by mosquitoes, scoring 1 point for identifying at least one disease correctly without selecting any incorrect option.

### 2.5. Attitude Score

The attitude score was based on answers to three questions. Two associated introductory questions on the “presence of mosquitoes” and “what time of day is the mosquito presence greatest” were not incorporated in the scoring. The next question asked for a rating of the nuisance from mosquitoes in the area of the respondent’s residence, with rating options from 0 (none) to 4 (very high). Respondents who chose options 2 (medium) to 4 (very high) were given a score of 1, while other options were scored 0. The second scored question asked the respondents if they considered the mosquito problem important for health, because of nuisance, or both. Respondents who chose the option “both” were given a score of 1. The third question in this group asked whether the respondents undertook any extra actions to control mosquitoes in their residences; if they answered yes, they were given a score of 1. For the refugees and migrants only, an extra question was asked on how important the mosquito problem was in their country of origin, with rating options from 0 (none) to 4 (very high). Respondents who chose options 2 (medium) to 4 (very high), were given a score of 1, while other options were scored 0.

### 2.6. Perception Score

The perception score was based on the response to “who is responsible for controlling the problem of mosquitoes?”. Responses “Camp/State authorities” and “myself” were scored 0. The response “both” was given a score of 1.

### 2.7. Entomological Surveillance

The composition of mosquito fauna in the Open Accommodation Refugee Camp at Malakasa, was investigated by the monitoring system of two BG-sentinel traps (BGs) (Biogents AG, Regensburg, Germany) baited with CO2 and BG-lure [29]. Due to limited access samples were collected every 10–15 days. All collected mosquito samples were transferred in chilled containers containing dry ice to the laboratory for morphological identification.

## 3. Results

### 3.1. Basic Description of Samples

The basic demographic description of the samples is shown in Table 1. The majority (63%) of respondents in the Malakasa camp was male, compared to 56% in Oropos and 45% in Moschato-Tavros. The camp respondents tended to be much younger than other respondents, with 62.0% below 35 years of age, compared to 27.1% in Oropos and 10.8% in Moschato-Tavros. The mean age was 46.5 (SD 16.1) in Oropos and 51.4 (SD 13.1) in Moschato-Tavros, but only 30.2 years (SD 10.0) in the camp.

Most of the camp respondents (77.0%) were living there with accompanying persons, 38.0% with four or more such persons. Exactly half of the camp respondents had resided there for less than 1 year, including 30.0% who had been there for less than 6 months. To a large extent this indicates the temporary nature of camp residence, which might have a bearing on the quality of the survey’s results in regard to long term consideration of KAP in this context. In addition, most of the respondents (60.0%) reported that the mosquito problem in their country of origin was of no or low importance (not shown in tables). Educational level was not ascertained in the camp, also due to the fact that most of the camps’ residents were coming from different non-homogenous cultural and educational contexts; the level was quite high in Oropos, where 42.4% had post-secondary education, less so in Moschato-Tavros (29.2%). Close to half of the respondents had children in the household. Although many houses in Oropos are second homes, nearly two-thirds of respondents (64%) said that in fact this was their permanent residence (not shown in tables).

### 3.2. Mosquito-Related Questions Compared between Samples

Table 2 presents responses to basic questions regarding the presence of mosquitoes and the nuisance that they cause. The percentage of respondents that rated the presence of mosquitoes as “very high” was 65.0% in the Camp, compared to only 22.0% in the municipality of Oropos and 22.6% in Moschato-Tavros (*p* < 0.001). This very large difference may partly be attributable to the fact that migrants live in worse conditions, including tents. Similarly, the much more frequent rating of the mosquito nuisance as “very high” in the camp—54%, compared to 14% in Oropos and 21% in Moschato-Tavros (*p* < 0.001)—may be associated with living conditions. The time of the day that the mosquito problem is greatest was said to be the nighttime by a much higher percentage (66%) of respondents from the camp and the neighbouring area of Oropos (70.0%) than in Moschato-Tavros (46.5%) (*p* = 0.003).

### 3.3. Knowledge and Attitude Responses

As shown in Table 3, none of the camp residents answered all three “Knowledge” questions correctly, and only 10–11% of the citizens of the municipality of Moschato-Tavros and Oropos (*p* = 0.003). In fact, only three out of the 100 respondents from the camp could give any answer to the question about which sex of mosquito bites. Corresponding to the lowest percentages of correct responses across the three samples, more respondents from the camp had a knowledge score of zero (38%), compared to 27% in Moschato-Tavros and 22% in Oropos.

One of the key questions in the “Attitudes section” was whether the respondents took extra measures for the control of mosquitoes, beyond the action taken by authorities. Whereas only about half of the camp respondents (53%) were taking extra measures, most of the remainder was doing so: 94% in Oropos and 92% in Moschato-Tavros (*p* < 0.001; data not shown). In addition, it should be noted that the percentage of respondents having replied correctly to all three “Attitudes” question is similar for all three sample areas ranging from 45% in the camp area to 48% in the other two urban and peri-urban areas (Table 4). On the question of whether mosquitoes present a health problem, a nuisance, or both (Table 5), the majority (88%) of residents of the camp replied that it is a health problem or both, the same percentage as in Oropos, but lower than the 96% in Moschato-Tavros (*p* = 0.025).

### 3.4. Perception Responses

As shown in Table 5, 39% of the camp’s respondents indicated that the authorities are solely responsible for dealing with the mosquito problem, a higher percentage than in Oropos (27%) and Moschato-Tavros (18%) (*p* = 0.001). In addition, on an additional public policy question, not included in the KAP scoring, on whether the public actions are sufficient for the mosquito problem, 20% of the camp’s respondents replied positively, compared to only 8.2% in the municipality of Oropos and 45.3% in Moschato-Tavros (*p* < 0.001). All answers being below 50%, they indicate a form of protest vote regarding authorities’ actions, each one from a different perspective.

In the second survey round in the Malakasa Camp, the correct response “both” to the question on responsibility for dealing with the mosquito problem was given by 78%, compared to 61% in the first survey (*p* = 0.024). This indicates a significant impact of the small-scale information session on KAP within the camp.

### 3.5. Mosquito Fauna Identification

A total of 936 (937♀, 17♂) adult mosquitoes were collected in all traps of the Malakasa camp from June 2020 to February 2021. The implemented entomological survey revealed the presence of three mosquito species. In Figure 3 the seasonal abundance of *Cx. pipiens s.l.* and *Ae. albopictus* females is presented. These two species were the only mosquitoes of medical importance collected in the camp premises. Other species collected, during the whole period, were *Ae. caspius* (10 adults) and *Culiseta longiareolata* (18 adults). Entomological surveillance revealed that the main collected species were *Cx. pipiens s.l.* (866♀, 15♂), *Ae. albopictus* (55♀, 0♂), *Cs. longiareolata* (16♀, 2♂) and *Ae. caspius* (10♀, 0♂), which is in accordance with previous studies [29,30]. The high abundance of *Cx. pipiens s.l.* explains the high percentage in responses related to mosquito nuisance during the night in Malakasa camp, specifically rated at 66% in the question “What time of day is the mosquito problem greatest?” of Table 2.

## 4. Discussion

The results of the current study are timely in regard to the humanitarian emergency of intensified migration flows in the Mediterranean, as well as the increased epidemiological risks associated with the increased presence of *Aedes* mosquitoes in this region. Mosquito-borne diseases have not been recorded in refugees camp in Greece, but other health threats related mostly with overcrowding and insufficient sanitation [31,32]. According to a recent study, important disease vectors and pathogens have been identified in the refugee camps in Greece, a fact that indicates an increased public health risk for transmission of diseases [33]. Therefore, the results of the current study can provide useful input into public policy decisions on the usefulness of community-based approaches for the control of the mosquito problem under different socio-economic settings.

It is possible that some of the responses reported in this survey have been affected by the special socio-demographic characteristics of the respondents from the migrant camp. As already noted, half of them had been a camp resident there for less than 1 year and the majority reported that the mosquito problem in their country of origin was none or low. It should be noted that, based on the records of the camp’s authorities, the great percentage of the residents’ origin at the time of the survey was Afghanistan, with fewer residents coming from Syria, Iraq and Iran. Most of the camp’s respondents reported that the mosquito problem associated with their country and area of origin was not severe, and it should be noted that malaria was the most commonly known disease among the respondents. In addition, their exposure to displacement and other major health threats could have generally affected the rating of the mosquito-related problems. This can certainly be regarded as a major factor which should be taken into account in designing effective community-based approaches and prioritizing the associated public health risk according to epidemiological and vector surveillance data [34].

Together, findings suggest that there might be substantial difficulty for those non-fully integrated subjects to internalize and understand mosquito-related problems in their transitory accommodation areas. Specifically, in regard to respondents’ answer to the “Presence of mosquitoes” it is worth noting that the percentage of respondents replying “very high” was 65.5% in the Camp, but only 22.0% elsewhere. This percentage is also attributable to the rate of migrants’ living in very basic and often inadequate conditions, such as tents. Similarly, the “Rating of nuisance” was reported as “very high” by over half of the Camp’s residents, but much fewer elsewhere. Results also indicate the large need for enhancing integrated approaches to community and awareness campaigns within and outside the context of the migration settlements, as the overall recorded levels of knowledge, attitudes and perception could be significantly improved.

In Greece, *Aedes* invasive mosquito species are aggressive day-biting mosquitoes while native species are active during the night. Therefore, in the question for the time of day that the mosquito problem is greatest, “both” includes native and invasive species, and was used to identify the presence of *Ae. albopictus* in a certain area [25,28,35]. Previous studies showed that *Aedes albopictus* presence is higher in urban areas of the Region of Attica [36].

The present study highlights the need for public investment in increased epidemiological and vector surveillance, as well as adequate infrastructure to ensure the elimination of vector breeding sites. The findings of the current study, are in accordance with other conducted studies, concerning a lower level of information and awareness of refugees on the problem of mosquito and household control practices [37]. On the other hand, in a study conducted in Greece, it was recorded that migrant workers showed higher levels of knowledge for malaria transmission, a fact that may also be possibly associated with their social status as workers. However, migrants’ and refugees’ housing conditions, as well as knowledge and cultural practices, all appear to be significant factors affecting the spread of vectors and their associated diseases. It should be noted that current findings also indicate the need to enhance the information and awareness campaigns, both within and outside the migrants’ settlements, for increasing the level of community engagement in crucial public health issues. In order to ensure inclusive whole-of-society public health interventions, it is necessary to apply clear scientific approaches on the risks and causes of the spread of similar diseases without marginalizing certain social groups, such as migrants and refugees [38].

The current study had several limitations. Certain biases and limitations were associated with the reporting of Knowledge, Attitudes and Perceptions of migrants in the current study. It should be noted that to a certain extent the high rating of nuisance and presence of mosquitoes, as well as of the widespread belief that the camp authorities were responsible for the mosquito problem, could be interpreted as a form of “protest” by the migrants and refugees against their living conditions, as well as the lack of social integration. Interviewers made an effort to repeat the questions and make sure that the answers were not a result of the refugees’ despair or anger and that they were addressed to the specific question posed. An additional limitation is that the quality of the collected responses may have been adversely affected using translation and even the comprehension of the information presented. In this case a further explanation was provided by interviewers to the specific questions, trying not to bias the level of responses. Furthermore, in certain cases the validity of responses could be questionable, mainly concerning information on the age or number of accompanying members due to reluctance to answer questions related to in-camp administrative procedures, despite assurances of protection of personal data.

It is essential for countries facing the challenge of intense migrant flows to ensure public health preparedness in the temporary settlements of migrants to certain risks including the spread of mosquito-borne diseases. The latest evidence indicates a need for strengthening preparedness at migrant centres specifically in regard to enhancing human resources, access to medicines and vaccines, adequate physical infrastructure and sanitation, health information and financing for controlling and mitigating different public health risks within and outside the context of migration centres [39]. It should be clear that studies similar to the one conducted here should help formulate public health approaches with a humanitarian responsibility, while at the same time recognize and call for the need of a renewed political commitment needed to address the migrant crisis in Europe and globally [40].

## 5. Conclusions

The current study highlights the need to design optimal public health policies considering the societal inclusion of migrants while ensuring public health security within and outside temporary migratory settlements. The poor level of knowledge indicates the need to intensify the provision of information to every subgroup of the population with particular emphasis on the group of migrants. On the other hand, it should be highlighted that, specifically for the group of migrants, relevant public health policies cannot be efficient if the necessary infrastructure and conditions are not available to support their implementation. Public health preparedness is essential for tackling upcoming challenges related to population movement and displacement due to climate change, political instability and the exposure of vulnerable populations to the risk of mosquito-borne diseases [41]. This type of study can be informative of the disparities existing in the levels of knowledge for certain public health issues as well as the feasibility of certain approaches, such as community-based approaches, for alleviating health-related issues such as mosquito-borne diseases, while at the same time public health policies should ensure social equity, leaving no one behind and offering equal access to health.

## Figures and Tables

**Figure 1 ijerph-19-16900-f001:**
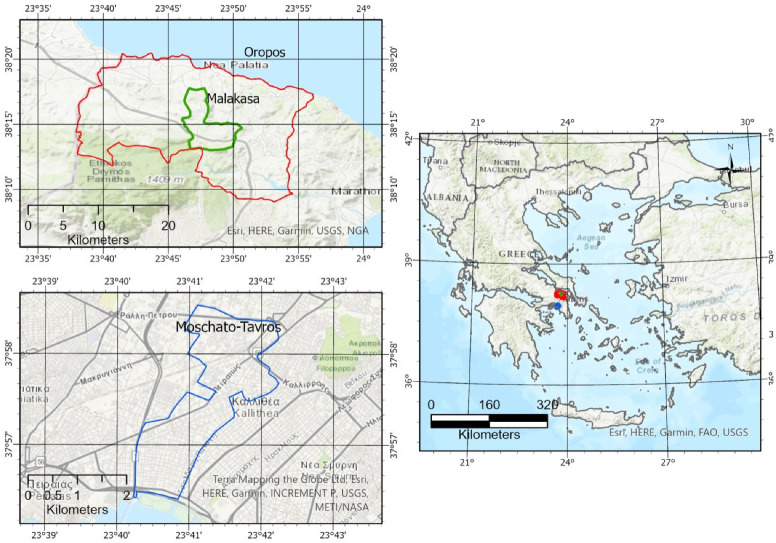
Map of the survey areas: Red frame: Territorial borders of the Oropos Municipality; Green frame: Territorial borders of the Malakasa community; Blue frame: Territorial borders of the Moschato-Tavros municipality.

**Figure 2 ijerph-19-16900-f002:**
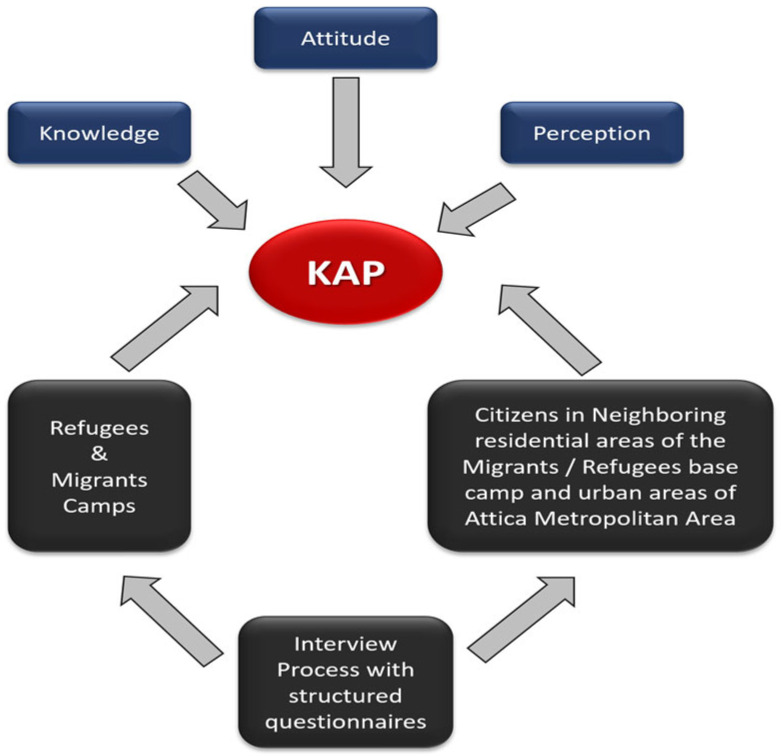
The implementation of KAP approaches in different citizenship contexts.

**Figure 3 ijerph-19-16900-f003:**
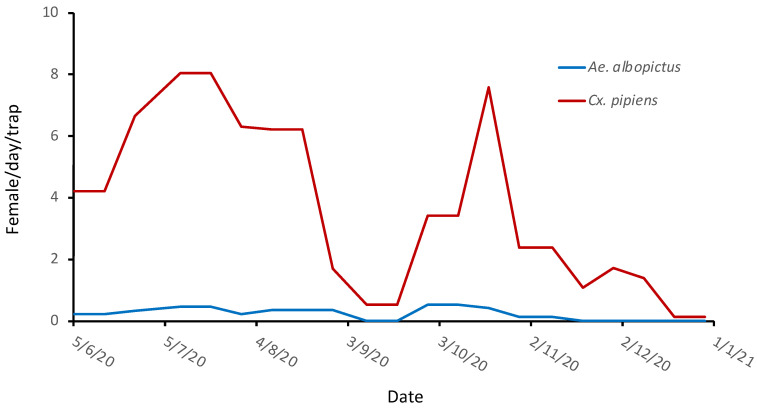
Number of females of *Cx. pipiens s.l.* and *Ae. albopictus* adults per day captured in the Malakasa migration camp per trap.

**Table 1 ijerph-19-16900-t001:** Sample characteristics in the three survey areas.

		Malakasa Migrant Camp		Urban Athens (Moschato Municipality)		Peri-Urban Neighboring with Camp (Oropos Municipality)	
		n	%	n	%	n	%
Sex	Male	63	63.0	87	55.8	21	44.7
	Female	37	37.0	69	44.2	26	55.3
Age	<25	24	24.0	0	0.0	2	4.2
	25–34	38	38.0	17	10.8	11	22.9
	35–44	24	24.0	33	21.0	13	27.1
	45–54	10	10.0	40	25.5	6	12.5
	55–64	4	4.0	41	26.1	11	22.9
	65+			26	16.6	5	10.4
Education Level	Primary			13	8.2	9	18.8
	Secondary/High School			78	49.4	25	52.1
	Higher/University			67	42.4	14	29.2
Accompanying persons	None	23	23.0				
	1	8	8.0				
	2	16	16.0				
	3	15	15.0				
	4	24	24.0				
	5+	14	14.0				
Children in the house	Yes			89	56.0	23	46.0
	No			70	44.0	27	54.0

**Table 2 ijerph-19-16900-t002:** Responses to basic questions related to mosquito presence and nuisance.

		Malakasa Migration Camp		Urban Athens (Moschato-Tavros Municipality)		Peri-Urban Neighboring with Camp (Oropos Municipality)	
		n	%	n	%	n	%
How high do you rate the presence of mosquitoes in your area of residence?	None	0	0.0	7	4.4	0	0
	Low	1	1.0	22	13.8	6	12.0
	Medium	12	12.0	51	32.1	18	36.0
	High	22	22.0	43	27.0	15	30.0
	Very High	65	65.0	36	22.6	11	22.0
What time of day is the mosquito problem greatest?	Day	1	1.0	7	4.4	0	0.0
	Night	66	66.0	74	46.5	35	70.0
	Both	33	33.0	78	49.1	15	30.0
How high do you rate the nuisance from mosquitoes in your area of residence?	None	0	0.0	12	7.5	0	0.0
	Low	2	2.0	25	15.7	5	10.0
	Medium	12	12.0	40	25.2	13	26.0
	High	32	32.0	49	30.8	25	50.0
	Very High	54	54.0	33	20.8	7	14.0
Do you know which mosquitoes bite?	Male	0	0.0	5	3.1	2	4.0
	Female	1	1.0	33	20.8	9	18.0
	Both	2	2.0	33	20.8	28	56.0
	Do not Know	97	97.0	88	55.3	11	22.0

**Table 3 ijerph-19-16900-t003:** Knowledge score: correct answers to Knowledge Questions.

			Camp	Urban Athens (Moschato Municipality)	Peri-Urban Neighboring with Camp (Oropos Municipality)
Knowledge Score	0	n	38	43	11
		%	38.0%	27.0%	22.0%
	1	n	50	63	23
		%	50.0%	39.6%	46.0%
	2	n	12	35	11
		%	12.0%	22.0%	22.0%
	3	n	0	18	5
		%	0.0%	11.3%	10.0%

**Table 4 ijerph-19-16900-t004:** Attitude score: Correct answers to Attitude Questions.

			Malakasa Migration Camp	Urban Athens (Moschato Municipality)	Peri-Urban Neighboring with Camp (Oropos Municipality)
Attitudes	0	n	1	1	0
		%	1.0%	0.6%	0.0%
	1	n	16	12	3
		%	16.0%	7.5%	6.0%
	2	n	38	69	23
		%	38.0%	43.4%	46.0%
	3	n	45	77	24
		%	45.0%	48.4%	48.0%

**Table 5 ijerph-19-16900-t005:** Replies to questions related to public policy in the three sample areas.

		Sample					
		Malakasa Migration Camp		Urban Athens (Moschato Municipality)		Peri-Urban Neighboring with Camp (Oropos Municipality)	
		n	%	n	%	n	%
Do you consider the mosquitoes problem important for	Nuisance	12	12.0	6	3.8	6	12.2
	Health	8	8.0	19	11.9	7	14.3
	Both	80	80.0	134	84.3	36	73.5
Are public control actions sufficient?	Yes	20	20.0	72	45.3	4	8.2
	No	80	80.0	87	54.7	45	91.8
Who is responsible for controlling the problem of mosquitoes?	Camp/State Authorities	39	39.0	29	18.2	13	27.1
	Myself	0	0.0	1	0.6	2	4.2
	Both	61	61.0	129	81.1	33	68.8

## Data Availability

All data are available in the manuscript.

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
