# Peer review of "Knowledge, Attitudes and Perception of Mosquito Control in Different Citizenship Regimes within and Surrounding the Malakasa Open Accommodation Refugee Camp in Athens, Greece"

_ijerph, 2022, doi:10.3390/ijerph192416900_

Round 1

Reviewer 1 Report

I believe this a good example on how important it is to investigate the different levels of knowledge to implement community-based and mosquito vector control interventions based on the attitudes and perceptions of the target community. I totally agree with the top highlight of the study, that evidences the role of the public health preparedness in tackling the challenges related to vector-borne diseases in a social scenario of human displacement due a variety of factors, including climate chance and political instability.

Overall, the article is clear, well written and the sections (introduction, material and methods...) discriminated. The methods are well described, however I belive that more information of the target areas could be provided. The results are presented only in tables, which are comprehensible, but not easy to read. This could eventually be improved using some graphical presentation, and the article would benefit. In the discussion is important that the authors identify and discuss the limitations of the study. References are current and appropriate for the study.

Please check the uploaded manuscript with my corrections and comments (yellow highlighted).

Major changes/ comments:

L49-50 refer that Aedes aegypti is established in madeira, Portugal, since 2005.In 2012 a dengue outbreak ocurred, the first in european region since 1920; 

L50 consider the recently introduction of aegypti in Cyprus

L68 Can you give examples or describe better?

L94 Is it possible to have more data on the area and number of inhabitants in the target areas?

Table. 1 Why is not present data on the education level of the migrant community?

L207 Reference or data on vector bionomics?

Table 2. In your opinion the high number for Both for the question What time of the day is the mosquito problem greatest? can be an indicator for the presence of Aedes albopictus mosquitoes? Why?

Author contributions is not filled

Informed consent not filled. Does informed consent applies?

Conflicts of interest not filled

Author Response

Overall, the article is clear, well written and the sections (introduction, material and methods...) discriminated. The methods are well described, however I belive that more information of the target areas could be provided. 

Thank you so much, highly appreciated

The results are presented only in tables, which are comprehensible, but not easy to read. This could eventually be improved using some graphical presentation, and the article would benefit.

Thank you so much for this comment; We have added another figure on the entomological translation of the results also based on available surveillance data and enriched accordingly the results section.

In the discussion is important that the authors identify and discuss the limitations of the study.

Study’s limitations already included in the Discussion. Nevertheless, we collected and moved them at the end of the Discussion.

L49-50 refer that Aedes aegypti is established in madeira, Portugal, since 2005.In 2012 a dengue outbreak ocurred, the first in european region since 1920;

L50 consider the recently introduction of aegypti in Cyprus

Thank you so much for this comment. Indeed, it is important to include Madeira’s outbreak and Ae. aegypti recent invasion in Cyprus and we accordingly did in the introduction section.

L68 Can you give examples or describe better?

L68 of the original manuscript has been revised for a better description and examples.

L94 Is it possible to have more data on the area and number of inhabitants in the target areas?

  Data on area and number of inhabitants added, as requested.

Table. 1 Why is not present data on the education level of the migrant community?

As also added to text, Educational level was not ascertained in the camp “also due to the fact that most of the camps’ residents were coming from different non homogenous cultural and educational contexts”

L207 Reference or data on vector bionomics?

This conclusion was moved in the Discussion. This is based on our surveillance data in Region of Attica (not all published yet). We revised the text, and a recent reference was included.

Table 2. In your opinion the high number for Both for the question What time of the day is the mosquito problem greatest? can be an indicator for the presence of Aedes albopictus mosquitoes? Why?

Indeed, this is always a good indicator for the presence of day biting mosquitoes since in Greece, native species are active during the night. Therefore, in the question for the time of day that the mosquito problem is greatest, “Both” includes native and invasive species and used to identify the presence of Ae. albopictus in a certain area.

A text and relevant references were included in the Discussion.

Author contributions is not filled

Informed consent not filled. Does informed consent applies?

Conflicts of interest not filled

The relevant information has been included in the manuscript.  

Reviewer 2 Report

I find the paper well written and ready for publication. Congratulations! A minor issue for me is FIGURE 2. It takes up a lot of space but does not convey any significant additional insights. Maybe you can refine this a bit.

3.2 Have you considered the lack of air con in camps as opposed to a (potential) availability within cities / the municipalities?!

Author Response

I find the paper well written and ready for publication. Congratulations!

Thank you so much, very highly appreciated.

A minor issue for me is FIGURE 2. It takes up a lot of space but does not convey any significant additional insights. Maybe you can refine this

Figure 2 has been edited to cover a smaller space on the manuscript.

3.2 Have you considered the lack of air con in camps as opposed to a (potential) availability within cities / the municipalities?!

Thank you so much for this comment; As highlighted within the text This very large difference may partly be attributable to the fact that migrants live in worse conditions, including tents.,  there has been a holistic consideration of the living conditions difference among the different population; In addition details about the living conditions in the camps have been provided in text: “The questionnaire survey of migrants and refugees was carried out in the Open Accommodation Refugee Camp at Malakasa, located on the geographical border of the Αthens area (Figure 1). There were 264 prefabricated accommodation containers and 29 apartments each hosting two families in separate rooms with common Water, Sanitation and Hygiene (WASH) Facilities (shower/bath), while a number of newly arrived migrants reside on temporary settlements including tents. Also, the Hellenic Water Company (EYDAP) provides untreated water and suitable treatment systems have been installed in order to comply with the relevant EU and national standards.”

Reviewer 3 Report

"Knowledge, Attitudes and Perception of mosquito control in different citizenship regimes within and surrounding the Malakasa Open Accommodation Refugee Camp in Athens, Greece", by Kolimenakis et al., presents a relevant topic as it explores the awareness of mosquito-borne diseases and the importance of mosquito control by vulnerable population and discusses its importance in the public health context.

Overall, the text is well-written and the information is clear. I have only a few comments/doubts:

- Across the text, the species/genus mentioned must be put in italics;

- Introduction

* Lines 46-61: I believe information about the distribution of mosquito species and related diseases in Greece could also be provided. There are examples from Italy, France, Croatia... but none from Greece.

*Are there any data about the prevalence of mosquito species / mosquito-borne diseases in refugee camps and/or refugee camps in Greece? I think it would be essential to contextualize before the final paragraph.

Materials and Methods

* I did not recognize the mention of signing any terms of consent/terms of agreement by the participants of the study. Was the research submitted for approval by an ethics committee? What legal/ethical precautions did the authors take, considering the subjects of research?

*Lines 125-126: "The structured questionnaire was designed on the basis of a small-scale pilot study by personal interviews among migrants and refugees residing in the camp". What pilot study is this? How it was elaborated? In which data it was based? How did the researchers responsible for its elaboration determined the questions that compose it?

-Results

*Lines 203-204: "The time of the day that the mosquito problem is greater was said to be the nighttime(...)". Considering Aedes mosquitoes are generally day-biting and most active during daylight hours, would not the previous affirmation suggest other genera? Culex sp, for example?

In fact, only Aedes is mentioned in the text. But as there are mentions of diseases not related to Aedes sp, such as malaria, would not be other important species in the areas evaluated?

 That is why becomes important to set more details about the geographical distribution of mosquito species and the epidemiology of related diseases in the context of the area explored in the study. Also, it would be possible to know which pathogens transmitted by mosquitoes affect the most people living in refugee camps.

Lines 217-219: "The poor level of knowledge indicates the need to intensify the provision of information to every subgroup of the population with particular emphasis on the group of migrants." 

Considering the perspective of mosquito control, in the situation in which those populations find themselves - with more basic and urgent needs to fulfill - what would be the efficacy of providing information but not necessarily providing the conditions to apply it?

It seems they have a good score in the "Attitude" section. If the knowledge increases, but the conditions for its practical application are not present, is there a real possibility to increase the attitude? 

It might be one possible interpretation, but it seems the lack of knowledge is the main reason for not sustaining mosquito control strategies, when are the living conditions that the migrant groups face that makes it a challenge.

- Discussion:

I believe the discussion is well-written, concise, and clear. The limitations of the study are well pointed out by the authors.

Author Response

Overall, the text is well-written and the information is clear.

Thank you so much, appreciated.

Across the text, the species/genus mentioned must be put in italics;

Corrected as requested.

Lines 46-61: I believe information about the distribution of mosquito species and related diseases in Greece could also be provided. There are examples from Italy, France, Croatia... but none from Greece. 

Since 2010 several outbreaks of West Nile virus infection occurred in Greece. Fortunately, there is no previous experience with mosquito–borne disease related to Aedes species, such as Dengue, Zika and Chikungunya. Furthermore, since 2009 a number of malaria cases have been recorded, mainly as sporadic introduced cases and in 2011-2012 in clusters.

Text and references added in the Introduction.

Are there any data about the prevalence of mosquito species / mosquito-borne diseases in refugee camps and/or refugee camps in Greece? I think it would be essential to contextualize before the final paragraph

Concerning refugees’ camps in Greece no mosquito-borne diseases have been recorded and all the major health care problems are related mostly with overcrowding and insufficient sanitation.

Text and references added in the Discussion.

I did not recognize the mention of signing any terms of consent/terms of agreement by the participants of the study. Was the research submitted for approval by an ethics committee? What legal/ethical precautions did the authors take, considering the subjects of research?

The current study was implemented in accordance with the Ethics Code for Research and the study was reviewed and authorized by the Hellenic Ministry of Migration and Asylum.

Considering the provision of the blank copy of the informed consent used, please note that at the beginning of each interview an introductory text on the scope of the interview was read on each subject to be interviewed, also requesting their consent to participate. The English translation of the questionnaire is provided for your kind information.

In addition, please also note that as stated in the manuscript:

"Most of the camp residents are Afghan nationals and interviews were conducted in their language by an interviewer accompanied by a translator who was provided by the International Organization of Migration office operating within the camp. Interviews were conducted on the camp premises, mostly outside the respondent’s residence, and the sample was selected based on the availability and willingness of each questioned subject to participate in the interview."

Lines 125-126: The structured questionnaire was designed on the basis of a small-scale pilot study by personal interviews among migrants and refugees residing in the camp". What pilot study is this? How it was elaborated? In which data it was based? How did the researchers responsible for its elaboration determined the questions that compose it?

Thank you so much for this important comment, the following lines were added:

“The structure of the questionnaire was based on previous similar conducted studies in the Greek territory, which was adapted to the needs of this survey by being pilot tested for its efficiency and feasibility within the camp’s context. Based on the pilot test, conducted to a small sample within the Malakasa camp with the support of International Organization of Migration office translators, the final set of questions was determined.”

Lines 203-204: "The time of the day that the mosquito problem is greater was said to be the nighttime(...)". Considering Aedes mosquitoes are generally day-biting and most active during daylight hours, would not the previous affirmation suggest other genera? Culex sp, for example?

In fact, only Aedes is mentioned in the text. But as there are mentions of diseases not related to Aedes sp, such as malaria, would not be other important species in the areas evaluated?

That is why becomes important to set more details about the geographical distribution of mosquito species and the epidemiology of related diseases in the context of the area explored in the study. Also, it would be possible to know which pathogens transmitted by mosquitoes affect the most people living in refugee camps.

This conclusion was moved in the Discussion. We revised the text, and relevant references were included.

No malaria vectors were collected (Anopheles sp). We added data from the entomological surveillance in the camp. In new Figure 3 we present the seasonal abundance of Culex pipiens and Aedes albopictus females. These 2 species were the only mosquitoes of medical importance that collected in the camp. Other species collected, during the whole period, were Aedes caspius (10 adults) and Culiseta longiareolata (18 adults).

In Greece no mosquito-borne diseases have been recorded in camps and all the major health care problems are related mostly with overcrowding and insufficient sanitation.

Lines 217-219: "The poor level of knowledge indicates the need to intensify the provision of information to every subgroup of the population with particular emphasis on the group of migrants." 

Considering the perspective of mosquito control, in the situation in which those populations find themselves - with more basic and urgent needs to fulfill - what would be the efficacy of providing information but not necessarily providing the conditions to apply it?

It seems they have a good score in the "Attitude" section. If the knowledge increases, but the conditions for its practical application are not present, is there a real possibility to increase the attitude? 

It might be one possible interpretation, but it seems the lack of knowledge is the main reason for not sustaining mosquito control strategies, when are the living conditions that the migrant groups face that makes it a challenge.

That is a very important comment and thank you so much for highlighting it. We totally agree and have considered adding few lines in the conclusion: On the other hand, it should be highlighted that specifically for the group of migrants, relevant public health policies cannot be efficient if the necessary infrastructure and conditions are not available to support their implementation.” And “...while at the same time public health policies should ensure social equity, leaving no one behind and equal access to health.”

Discussion: I believe the discussion is well-written, concise, and clear. The limitations of the study are well pointed out by the authors.

Thank you so much, appreciated.